# Are International Units of Anti-HBs Antibodies Always Indicative of Hepatitis B Virus Neutralizing Activity?

**DOI:** 10.3390/vaccines11040791

**Published:** 2023-04-04

**Authors:** Yada Aronthippaitoon, Nathan Szerman, Nicole Ngo-Giang-Huong, Syria Laperche, Marie-Noelle Ungeheuer, Camille Sureau, Woottichai Khamduang, Catherine Gaudy-Graffin

**Affiliations:** 1Department of Medical Technology, Faculty of Associated Medical Sciences, Chiang Mai University, Chiang Mai 50100, Thailand; 2LUCENT International Collaboration, Faculty of Associated Medical Sciences, Chiang Mai University, Chiang Mai 50100, Thailand; 3Laboratoire de Bactériologie-Virologie-Hygiène, CHRU, Université of Tours, INSERM U1259, 37044 Tours, France; 4Institut de Recherche pour le Développement (IRD), MIVEGEC, CNRS, Agropolis, University of Montpellier, 34394 Montpellier, France; 5Institut National de la Transfusion Sanguine, CNR Risques Infectieux Transfusionnels, 75015 Paris, France; 6Etablissement Français du Sang, La Plaine, 93218 Saint-Denis, France; 7Center for Translational Science, ICAReB, Institut Pasteur, 75015 Paris, France

**Keywords:** HBV vaccine, anti-HBs international units, anti-HBs neutralizing activity, HBV infectivity neutralization

## Abstract

Objective: Anti-HBs antibodies are elicited upon hepatitis B vaccination, and concentrations above 10 mIU/mL are considered protective. Our aim was to assess the relationship between IU/mL of anti-HBs and neutralization activity. Methods: Immunoglobulins G (IgGs) were purified from individuals who received a serum-derived vaccine (Group 1), a recombinant vaccine, Genevac-B or Engerix-B (Group 2), or who recovered from acute infection (Group 3). IgGs were tested for anti-HBs, anti-preS1, and anti-preS2 antibodies and for their neutralizing activity in an in vitro infection assay. Results: Anti-HBs IUs/mL value did not strictly correlate with neutralization activity. The Group 1 antibodies demonstrated a greater neutralizing activity than those of Group 2. Anti-preS1 antibodies were detected in Groups 1 and 3, and anti-preS2 in Group 1 and Group 2/Genhevac-B, but the contribution of anti-preS antibodies to neutralization could not be demonstrated. Virions bearing immune escape HBsAg variants were less susceptible to neutralization than wild-type virions. Conclusion. The level of anti-HBs antibodies in IUs is not sufficient to assess neutralizing activity. Consequently, (i) an in vitro neutralization assay should be included in the quality control procedures of antibody preparations intended for HB prophylaxis or immunotherapy, and (ii) a greater emphasis should be placed on ensuring that vaccine genotype/subtype matches with that of the circulating HBV.

## 1. Introduction

Hepatitis B virus (HBV) can cause acute and chronic liver diseases in humans. Worldwide, it is estimated that more than 250 million individuals are living with chronic HBV infection and, hence, are at risk of developing cirrhosis and hepatocellular carcinoma (HCC). HBV is an enveloped virus and the prototype of the *Hepadnaviridae* family [1]. Its DNA genome codes for three integral envelope proteins, which differ from each other by the size of their *N*-terminal ectodomain. They all bear the hepatitis B surface antigen (HBsAg) and are referred to as large, middle, and small HBsAg proteins (L-HBsAg, M-HBsAg, and S-HBsAg, respectively) [2]. Their function in the HBV replication cycle resides in coating the HBV nucleocapsid for the assembly of infectious HBV virions; occasionally, they also coat the ribonucleoprotein (RNP) of the Hepatitis Delta Virus (HDV), leading to the production of HDV virions [3]. In the bloodstream, the envelope proteins confer stability to HBV and HDV virions, and they carry out the early steps of viral entry into human hepatocytes by mediating an initial attachment of the virion to the cell surface of heparan sulfate proteoglycans (HSPGs) [4] and a subsequent high affinity binding to the sodium taurocholate co-transporting polypeptide (NTCP) [5]. Binding to NTCP is mediated by the *N*-terminal, the preS1 domain of L-HBsAg, whereas the prior attachment to HSPG is driven by the antigenic loop (AGL) of the HBsAg proteins, an amino acid sequence that bears the immunodominant HBsAg “a” determinant. The latter elicits the production of neutralizing anti-HBs antibodies in individuals who recover from acute infection or in response to hepatitis B (HB) vaccination [6].

The first developed HB-vaccine consisted of HBsAg subviral particles purified from plasma pools of HB chronic carriers, and as early as 1981, this plasma-derived vaccine was demonstrated to be highly efficacious in inducing anti-HBs antibodies and in preventing acute hepatitis B and asymptomatic infection. It became commercially available in 1982 [7,8]. Because of cost efficiency and safety concerns, second-generation vaccines were developed, consisting of recombinant HBsAg proteins expressed in yeast or CHO cells [9,10]. Both induced high levels of anti-HBs antibodies and protection against hepatitis B; they have since been used extensively in an attempt to eradicate HBV infection worldwide [11]. Forty years after its introduction, the HB vaccine thus appears to be one of the most efficacious and safe human vaccines; however, in a few particular settings, such as in infants born from infected mothers [12] or patients undergoing liver transplantation, HB vaccination may fail, suggesting that there is still room for improvement of the vaccine itself or vaccination strategy.

Currently, the levels of anti-HB antibodies, expressed in international units (IU), are used to monitor the response to vaccination and to qualify the HB immunoglobulin (HBIG) preparations that are used for post exposure prophylaxis, and anti-HBs IU/mL levels are assumed to correlate with protection from infection. The present study was undertaken to question this assumption, using an in vitro infection assay that can functionally characterize anti-HB antibodies for their capacity to neutralize virus infectivity, a requirement for antibodies elicited by a sterilizing vaccine.

## 2. Materials and Methods

### 2.1. Serum Samples

Serum samples were for panels of the repositories of Institut National de la Transfusion Sanguine (INTS) registered as French ministry of health INTS cohort DC-2019-3559, the ICAReB platform (CoSimmGEn and Diagmicoll cohorts) at the Pasteur Institute (Paris), and the laboratory of Virology at the Tours University Hospital, France (Occupational Medicine Services cohort). Adult individuals had received 3 doses of either (i) plasma-derived Hevac-B vaccine (N = 56), between 1985 and 1992 (Group 1); and (ii) recombinant Engerix-B (N = 14), at 2009–2017, or Genhevac-B vaccine (N = 19), at 2009–2016 (Group 2). Also included in this study were serum samples from individuals who recovered from HBV infection (N = 19), (Group 3). All samples were stored at −20 °C.

### 2.2. Monoclonal Antibodies (mAbs)

MAbs used as controls for neutralization assays include anti-preS1 F3525 and MA18/7 antibodies; anti-preS2 Q19/1; anti-HBs A1.2, A2.1, H166, and C20/02; and anti-HIV GP120 [13,14].

### 2.3. Purification of IgGs from Serum Samples

The purification of IgGs was performed using the Melon Gel IgG Purification Kit (ThermoFisher Scientific) according to the manufacturer’s protocol. A total of 50 μL of serum was diluted 1/10 in Melon gel purification buffer prior to loading on Melon gel IgG spin columns. Purified IgGs were adjusted to 2 mg/mL, using a NanoDrop spectrophotometer (Thermo Scientific) and controlled for quality by polyacrylamide gel electrophoresis (PAGE) and Coomassie blue staining.

### 2.4. Measurement of Anti-HBs, Anti-preS1, and Anti-preS2 Antibodies by ELISA

Anti-HBs antibody levels were measured using an enzyme-linked immunoassay (ELISA) (Monolisa Anti-HBs PLUS, Bio-Rad). Anti-preS1 and anti-preS2 antibodies were measured using an in-house ELISA in which each well of a 96-well MaxiSorp plate (Nunc) was coated with 2 μg of a mix of preS1 specific peptides (positions 1–37 and 83–106) or preS2 peptides (109–134, 122–140, and 125–148) in 100 μL of 0.05 M NaHCO3 (pH: 9.60). The preS1 and preS2 peptides’ sequences were specific to HBV genotype D ayw3 (Genebank: X85254) and genotype A adw2 (Genebank: AY128092). The plate was incubated at 4 °C overnight and then treated with 300 μL/well of blocking buffer (0.05 M NaHCO3 (pH: 9) and 10% fetal bovine serum (FBS)) for 1 h at 37 °C. The plate was washed 5 times with phosphate buffer saline (PBS), 0.1% Tween 20 (PBST). After washing, 100 μL/well of 1:2 dilutions of purified IgG samples in PBST, 10% FBS, was added, and the plate was incubated for 1 h at 37 °C. After 5 washes with PBST, 100 μL/well of HRP-conjugated anti-human antibodies in PBST, 10% FBS, was added. The plate was finally washed 5 times with PBST before 100 μL/well of substrate (Ultra TMB) was added for 20 min at RT. Standard curves were established using polyclonal anti-preS1 (R271) or anti-preS2 (R257) antibodies raised in rabbit against a preS1 peptide (preS aas 1–37 genotype D-ayw3) or a preS2 peptide (preS aas 109–133 genotype D ayw3), respectively [15]. Anti-preS1 and anti preS2 ELISA titers were expressed in μg equivalent of R271- or R272- IgG positive controls, respectively.

### 2.5. Cell Culture and In Vitro HDV Infection Assay

Production and characterization of viral particles used for the in vitro infection assays were performed as previously described [16]. HDV RNA was quantified by Northern blotting. HBV envelope proteins were characterized by immunoblotting and quantified using an HBsAg ELISA kit (Wantai, China). Prior to inoculation, HDV preparations were normalized to 1 × 10^9^ ge/mL. Infection assays were performed as described [17,18], using the Huh-106 cell line. Cells were seeded in 96-well plates at 2.5 × 10^4^ cells per well. One day post-seeding, cells were exposed to HDV for 16 h at a multiplicity of infection (m.o.i) of 400 genome equivalent (ge). Post inoculation, culture medium was changed every 3 days, and cells were harvested at 9 days post inoculation (dpi) for measurement of intracellular HDAg proteins as readout of infection. The level of intracellular HDAg at 9 dpi is proportional to the percentage of infected cells, i.e., to the HDV titer in the inoculum (Appendix A).

### 2.6. Detection of Viral Nucleic Acids

At 9 dpi, total cellular RNA was purified using a GeneJET RNA Purification Kit (Thermo Fisher), according to the manufacturer’s protocol. The purified RNA was analyzed by Northern blotting, as described [16]. For detection of HDV RNA in the cell culture medium, RNA purification was performed using the QIAamp Viral RNA Kit (Qiagen), according to the manufacturer’s instructions. Viral RNA was then analyzed by Northern blotting. The detection of HBV DNA in 181 cell culture supernatants was carried out as described previously [16].

### 2.7. In Vitro Neutralization Assay

Purified IgG samples were normalized to 2 μg/μL before neutralization assays. Huh-106 cells were seeded in 96-well plates and 18 h post-seeding, and cells were exposed for 16–18 h to HDV virions (400 m.o.i.) that had been pre-incubated, or not, for 1 h at 37 °C with 1:2 dilutions of purified IgG. At 9 dpi, cells were harvested and lysed in 50 μL/well of 25 mM Tris-HCl (pH: 8.00), 1% NP40, 0.1% SDS, 0.5% sodium deoxycholate, and 150 mM NaCl. After a 2 min incubation at RT, 200 μL of 25 mM Tris- HCl (pH: 8.00), 2.5 mM MgCl_2_, and 25 unit/mL Benzonase (Merck) was added, and the mix was incubated at 37 °C for 1 h. Intracellular HDAg was then measured using a commercial HDAg ELISA (Diapro), following the manufacturer’s instructions. At the suboptimum m.o.i. of 400 ge, intracellular HDAg levels at 9 dpi are proportional to antibody-neutralizing activity (Appendix A).

### 2.8. Fluorescence Microscopy

The detection of intracellular HDAg at 9 dpi was achieved as described [18]. At 9 dpi, cell monolayers were fixed with 4% formaldehyde for 10 min at RT, washed with PBS, and incubated in the presence of an anti-HDAg antibody-positive human serum at 1:1000 dilution for 1 h at RT. Monolayers were then washed twice with PBS, incubated with Alexa Fluor-labeled goat anti-human secondary antibody (ThermoFisher) for 1 h, and mounted directly in Prolong gold antifade reagent (Molecular Probes) containing 1 μg/mL of 4′,6-diamidino-2-phenylindole 206 dihydrochloride (DAPI). Monolayers were examined using a Zeiss LSM 710 Meta confocal microscope.

### 2.9. Statistical Analysis

The correlation between anti-HBs antibody level and the 50% inhibitory concentration (IC50) in the neutralization assay was evaluated using Spearman’s rank correlation test. Comparison of median IC50 level between groups was performed using Mann–Whitney U test comparisons test. All statistical analysis and graphics were generated using GraphPad software 9.0 (GraphPad Software, Inc., San Diego, CA, USA). Statistical significance was considered when the *p*-value was <0.05.

## 3. Results

### 3.1. Neutralization of Virus Infectivity with Monoclonal Anti HBs Antibodies Depends upon HBsAg Subtype

As a preliminary experiment to our study, we performed an in vitro neutralization assay by using the HDV infection model described previously [18,19] and a selection of monoclonal antibodies (mAbs) directed to the preS1, preS2, or HBs antigens. MAbs were tested for their capacity to neutralize virions bearing envelope proteins of different HBV genotypes/subtypes. As shown in Appendix A, anti-S antibodies, previously described as directed to the HBsAg “a” determinant common to envelope proteins of all human HBV isolates, were not equally neutralizing across genotypes/subtypes. For example, MAbs A2.1 and C20/2 fully neutralized virions bearing genotype Aadw2 envelope proteins but poorly neutralized genotype E ayw4 HDV virions. In contrast, mAb 9709 neutralized A ayw1-HDV efficiently but was inactive against Badw2 HDV. This result prompted us to further investigate vaccine-derived anti-HB antibodies for their virus neutralizing activity.

### 3.2. Characterization of Antibodies Elicited by Hevac-B, Engerix, and Genhevac-B Vaccines or during Acute Infection

Anti-HBs antibody levels were measured in serum of (i) Hevac-B recipients (Group 1), (ii) Engerix or Genhevac-B recipients (Group 2), and (iii) individuals who recovered from acute infection (Group 3) (Figure 1). The median anti-HBs titer was 4571 mIU/mL (IQR: 164–12257 mIU/mL) in the Hevac-B group, 159 mIU/mL (IQR: 2–1154.0) in the Engerix-B, 1139 mIU/mL (IQR: 72–2408) in the Genhevac-244 B recipients, and 1249 mIU/mL (IQR: 941–2556) in the recovery group. Anti-preS1 antibodies were tested in the Hevac-B and recovery groups (Figure 1). The median anti-preS1 antibody level in Hevac-B recipients was 0.71 μg R271/mL (IQR: 0.43–1.05), which is higher than that of the recovered group (0.20 μg R271/mL; IQR, 0.11–0.51). In regard to anti-preS2 antibodies in Hevac-B and Genhevac recipients, the median level was 0.9 μg/mL, (IQR: 0.07–4.75) and 6.5 μg/mL (IQR: 2.7–16.9), respectively.

### 3.3. Establishment of a Quantitative Immunoassay for Measurement In Vitro HDV Infection

For measuring the neutralization activity of a large number of antibodies, we used a commercial HDAg ELISA to quantify the accumulation of HDAg proteins at 9 dpi in infected cells. We observed a linear dose–response for 100 < m.o.i. < 1000 (Appendix A). For the neutralization assay, serial 1:2 dilutions of IgG preparations were added to HDV virions prior to inoculation to Huh-106 cells at 400 ge m.o.i. Using anti-preS1 and anti-S IgG controls, we showed that intracellular HDAg at 9 dpi is inversely proportional to neutralization activity.

### 3.4. Correlation between Anti-HBs IgG ELISA and Neutralizing Activity

Of the 56 serum samples of the Hevac-B group tested by ELISA, 8 (H48, −39, −56, −42, −50, −43, −55, and −49) tested negative (<10 mIU/mL). However, when anti-HBs ELISA was carried out with purified IgGs, 14 more samples (H51, −52, −53, −42, −45, −47, −36, −46, -38, −54, −34, −44, −37, −41, and −40) tested negative as a result of the 1:10 dilution step in the IgG purification protocol (Figure 2A). The results of the neutralization assay (left histogram) are presented for all 56 samples and ranked by IC50 in μg IgG/mL in comparison to anti-HBs ELISA titers (Appendix A). The IC50 values are widely distributed from 1.7 to 200 μg/mL of IgG and not directly correlated to anti-HBs ELISA titers, expressed as mIU/μg of purified IgGs (right histogram). As expected, some samples, such as H 24 with a high anti-HBs value by ELISA, were highly neutralizing (low IC50). A few samples did not present any correlation between anti-HBs concentration and neutralization activity, such as H 33, characterized by a low anti-HBs ELISA titer and a high neutralizing activity, whereas other samples, such as H 17, were neutralizing despite a low anti-HBs level. Overall, our results show that anti-HBs ELISA titers in Hevac-B recipients are not fully indicative of a neutralization potency. The same conclusion can be made with regard to antibodies derived from individuals vaccinated with recombinant Engerix or Genhevac vaccines or present in patients who recovered from an acute infection (Figure 2A).

### 3.5. Correlation between Anti-preS Antibodies Titers and In Vitro Neutralizing Activity

All preparations of purified Hevac-B IgG were tested for the presence of anti-preS1 and anti-preS2 antibodies. As shown in Figure 2B, the levels of anti-preS1 or anti-preS2 antibodies did not correlate with HDV infectivity neutralization. However, samples such as H 33, characterized by low anti-HBs levels, may achieve neutralization through the activity of anti-preS2 antibodies. Surprisingly, a high concentration of anti-preS1-specific antibodies did not correlate with neutralization activity (see H 53, H 41, and H 29 samples). Overall, we did not observe a correlation between the neutralization of infectivity and the level of either anti-preS1 or anti-preS2 antibodies.

### 3.6. Neutralizing Activity of Anti-HBV Antibodies according to HBV Genotype of the Inoculum

We produced HDV virions bearing the envelope proteins of 10 different HBV isolates, genotype A-subtype adw2 (GenBank: MW357582), A-ayw1 (GenBank: MW357583), B-adw2 (GenBank: MW357584), B-ayw1 (GenBank: MW357585), C-adrq+ (GenBank: MW357586), D-ayw2 (GenBank: MW357587), D-ayw3 (GenBank: MW357588), E-ayw4 (GenBank: MW357589), F-adw4 (GenBank: MW357590), and G-adw2 (GenBank: MW357591). The amino acid sequences of the corresponding envelope proteins are presented in Appendix A. HDV preparations 1 to 10 were characterized as described (Figure 3) for the presence of HBV envelope proteins by immunoblot analysis, HBsAg ELISA, HDV RNA, and infectivity in Huh-106 cells after the normalization of the inoculum at 400 ge m.o.i. As shown in Figure 3c, infection could not be documented for Huh-106 cells exposed to virions specific for the A-adw2, C-adrq+, and F-adw4 HBsAg.

We then selected 10 IgG preparations that demonstrated various neutralizing activities, as reported in Figure 2A in (i) the Hevac-B group, (ii) the recombinant (Genhevac -B and Engerix) group, and (iii) the recovery group. All samples were normalized to their respective IC50 concentration and tested for neutralization activity against HDV-Aayw1, -B-adw2, -B-ayw1, -D-ayw2 and 3, -E-ayw4 and 9, and -G-adw2 at a 400 ge m.o.i. As shown in Figure 4, all antibodies achieved 50% inhibition of infection with the laboratory D-ayw3 prototype. Antibodies in the Hevac-B group demonstrated a broad neutralizing activity against HDV bearing a non-D-ayw3 HBsAg. Notably, sample H- 10 displayed a greater neutralizing activity against HDV-B-adw2, -B-ayw1, and -D-ayw3. Although samples from the recombinant group were more potent against HDV-Aayw1, they were less so against B-ayw1 and D-ayw2. In the recovery group, a better neutralizing activity was observed against the E-ayw4 group. All together, these results indicate that the neutralizing activity of vaccine-derived or naturally acquired anti-HBs antibodies varies according to the virus HBsAg subtype.

### 3.7. Neutralizing Activity of Anti-HBs Antibodies against Immune Escape D144A and G145R Variants

Antibodies listed in Figure 4 were then tested for their capacity to neutralize HDV virions bearing the main immune-escape HBsAg variants D144A and G145. Both vaccine-derived and infection-elicited anti-HBs antibodies were significantly impaired in neutralizing the two variants (Figure 5), whereas all types of antibodies could equally neutralize virions bearing the adw or ayw HBsAg.

## 4. Discussion

The efficacy of the HB vaccine at preventing hepatitis B transmission has been largely proven over the last four decades, especially in countries such as Taiwan and Thailand, where universal HB vaccination in infants has been implemented since the 1980s and 1990s, respectively [20,21,22]. This public health policy has had an impressive amount of success in lowering both the prevalence of hepatitis B and the incidence of HBV-related HCCs in the 2000s. However, failures of HB immunoprophylaxis can still occur in rare circumstances, such as in infants born to infected mothers and in the case of an HBsAg subtype mismatch between vaccine and circulating strain [12,23,24]. Whether vaccine failure could be due to a suboptimal compliance with the vaccination schedule, an inappropriate vaccine storage temperature, or to a poor immune response to vaccination in some individuals remains to be investigated.

In this study, we inquired about the functional aspect of vaccine-induced antibodies and precisely with regard to their capacity to neutralize virus infectivity. The capacity of an antibody to neutralize virus infectivity is only one aspect of the complex immune response to vaccination, but it is an essential characteristic of a sterilizing vaccine, which we aimed to document. Despite the limited number of samples for each group, our results were obtained with well-documented serum samples, and, overall, they show that whatever the immunogen, i.e., plasma-derived vaccine, recombinant vaccine, or natural infection, anti-HBs antibodies are produced at levels that do not necessarily translate into neutralization of infectivity; in other words, the sterilizing activity of anti-HBs antibodies varies according to the HBsAg subtype of the circulating virus, and it does not entirely correlate with the IU level.

Neutralization of virus infectivity can occur through different mechanisms, such as the aggregation of infectious virions, destabilization of the virion envelope, prevention of virion attachment to susceptible cells, or interference with a fusion mechanism. There is often a large variety of interactions between antibodies and neutralizing epitopes at the surface of a viral particle. At the surface of HBV or HDV virions, most anti-HBs antibodies bind to conformational epitopes of the HBsAg “a” determinant, which are more or less neutralizing, depending on the antigen–antibody affinity/avidity and the step at which antibodies interfere with the viral entry process [25,26]. Because the immunodominant “a” determinant is borne by an amino acid sequence that also bears an infectivity determinant [4,27], anti-HBs antibodies are likely to prevent virus attachment to the cell surface of heparan sulfate proteoglycans (HSPGs). Similarly, anti-preS1 antibodies present in the Hevac-B vaccine recipients may directly interfere with viral entry by preventing the interaction between the preS1 domain of the HBV/HDV envelope and the NTCP receptor at the human hepatocyte surface. To our knowledge, there is no indication that the HBV and HDV envelopes differ substantially from each other since both bear the pre-S1, pre-S2, and HBs antigens, and HBV and HDV virions can be captured by anti-preS1, anti-preS2, and anti-HBs antibodies. The preparations of HDV particles used in this study include the three HBV envelope proteins, L-, M-, and S-HBsAg. However, L-HBsAg accounts for approximately 25% of the total envelope proteins at the surface of HBV virions, and 5–6% at the surface of HDV virions (or HBV subviral particles). This difference would eventually have a marginal impact on the susceptibility to neutralization by anti-preS1 antibodies. Based on the capacity of anti-preS1 and anti preS2 antibodies to neutralize HBV infectivity [28], a third-generation HB vaccine that includes preS1 and preS2 antigens in addition to HBsAg was developed and shown in a prospective study to enhance protection in comparison to HBsAg-alone vaccines [29,30]. In our study, we show that the presence of preS antigens in the Hevac-B and Genevac-B vaccines has indeed led to the production of anti-preS antibodies. However, we could not document a correlation between anti-preS antibodies and the neutralization of infection in vitro. It is known that neutralizing antibodies may act after being internalized to interfere with the virus intracellular trafficking post-entry, but the characteristic of our assay based on the use of HDV as a reporter of infection did not allow for this investigation. However, the HDV based 96-well format of our neutralization assay was proven to be robust and convenient, with some room for sensitivity improvement that could be achieved by combining the use of lower m.o.i. with a more sensitive detection of intracellular HDV RNA by quantitative RT-PCR. We propose that such an assay be included in the quality-control procedures of HBIG or monoclonal antibody preparations that are intended for prophylaxis or immunotherapy. Since the neutralizing activity of vaccine-derived anti-HB antibodies varies according to the HBsAg subtype of circulating virus, passive and active HB immunoprophylaxis approaches could be improved by, for instance, ensuring that the HBsAg subtype of HB vaccine or HBIG matches that of the circulating HBV strain, especially in countries where mother-to-child transmission is still a concern. The latter suggestion could be efficiently implemented at the national health authority level if a wider variety of HB vaccines were made available. For the surveillance and prevention of HB vaccine failure, the assay could be conducted by virology reference laboratories. Overall, our results show that all vaccine-induced anti-HB antibodies are not created equal with regard to their sterilizing potency, a clear indication that protection against HBV infection is not solely predicted by serum anti-HBs’ antibody level.

## Figures and Tables

**Figure 1 vaccines-11-00791-f001:**
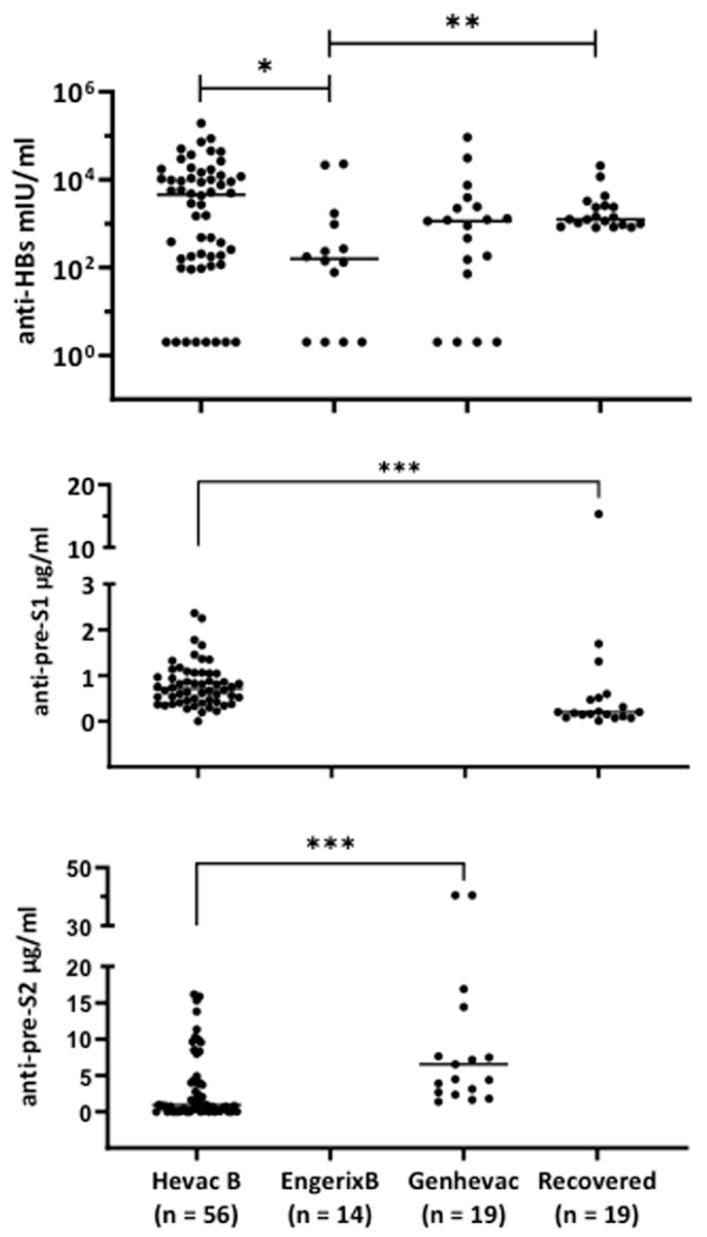
Anti-HBs, anti-preS1, and anti-preS2 antibodies’ detection in the plasma of individuals vaccinate with Hevac-B, Engerix-B, and Genhevac-B vaccines or after recovery from acute infection. Each dot corresponds to an individual subject. Anti-preS1 and anti-preS2 ELISA titers are expressed in equivalent of μg/mL of R271- or R272-IgG positive controls, respectively. * *p*-value < 0.05, **, *p*-value < 0.01, ***, *p*-value < 0.001.

**Figure 2 vaccines-11-00791-f002:**
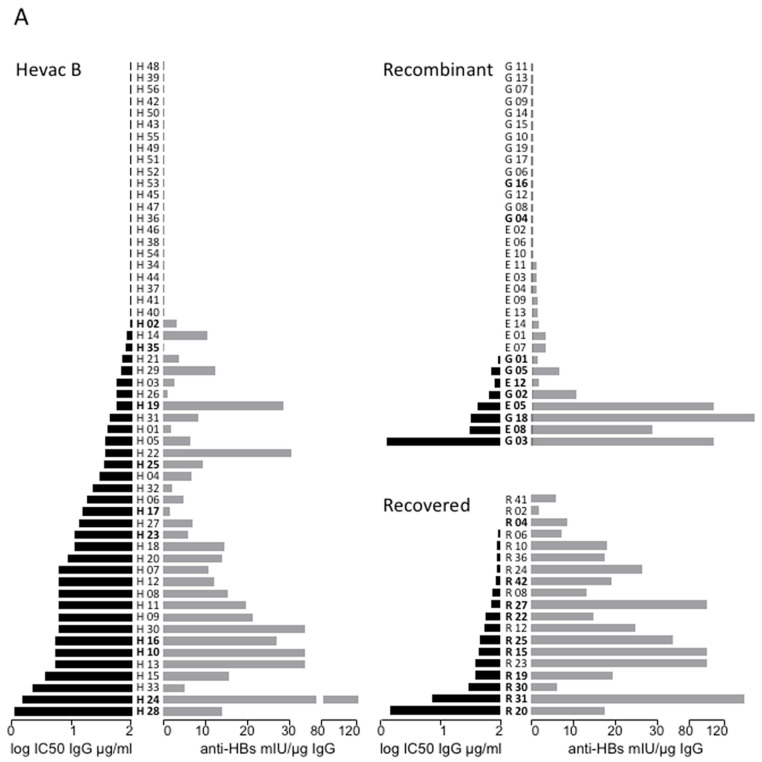
(**A**) Anti-HBs ELISA titers are not directly related to in vitro neutralization of HDV infectivity. Comparison of HDV neutralization activity (black histograms) of anti-HBs antibodies with their ELISA titers (gray histograms) in preparations of IgG purified from recipients of Hevac-B vaccine (H 01 to 56), Engerix-B vaccine (E 01 to 14), and Genehevac-B (G 01 to 19) and from individuals who recovered from infection (R 02 to 42). The IC50 was calculated using GraphPad Prism and expressed in μg of IgG/mL. Anti-HBs antibody titers are expressed in mIU/μg of purified IgG. (**B**) Comparison of HDV neutralization activity (black histograms) of Hevac-B elicited antibodies with the ELISA titers (gray histograms) specific for anti-HBs, anti-preS1, and anti-preS2 antibodies in IgG preparations. The IC50 was calculated using GraphPad Prism and expressed in μg of IgG per mL. Anti-HBs antibody titers are expressed in mIU/μg of purified IgG. Anti-preS1 and anti-preS2 ELISA titers are expressed in equivalent of μg of purified R271 and R272 IgGs per mL, respectively.

**Figure 3 vaccines-11-00791-f003:**
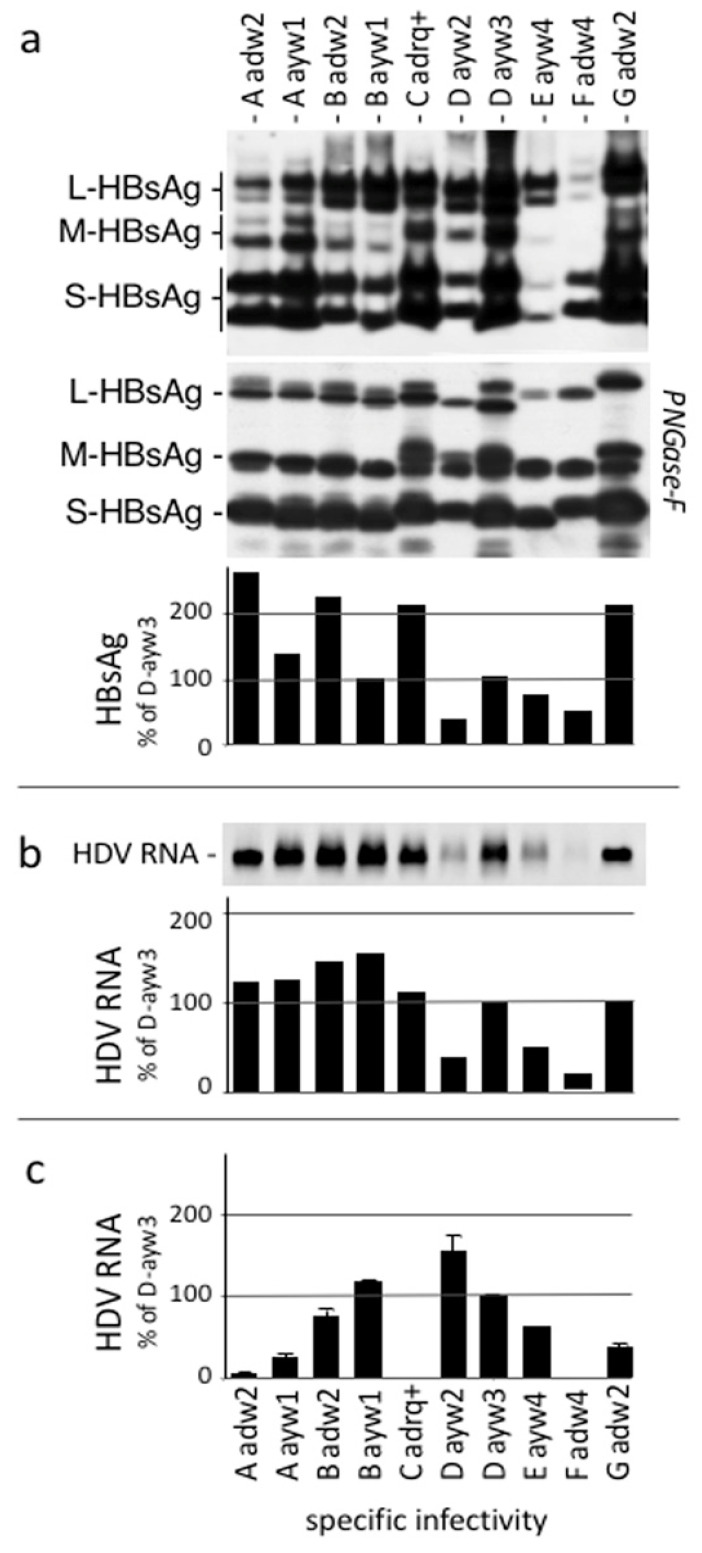
Characterization of HDV particles bearing envelope proteins of different HBV isolates. (**a**) HBV envelope proteins analyzed by Western blot (WB) analysis before and after incubation with PNGase F, as indicated, or by HBsAg-specific ELISA, and for HDV RNA by Northern blot analyses (**b**). Histograms show the relative amounts total HBsAg in each preparation (**a**) and of virion-associated HDV RNA (**b**). The glycosylated and non-glycosylated forms of the S-, M-, and L-HBsAg are indicated. Huh-106 cells were exposed to each preparation of HDV at a 400 m.o.i., and at 9dpi, cells were harvested, and cellular RNA was analyzed by Northern analysis (**c**). Quantification of HDV RNA signals was achieved using a phosphorimager. Histogram values are expressed as percentages of the wt value.

**Figure 4 vaccines-11-00791-f004:**
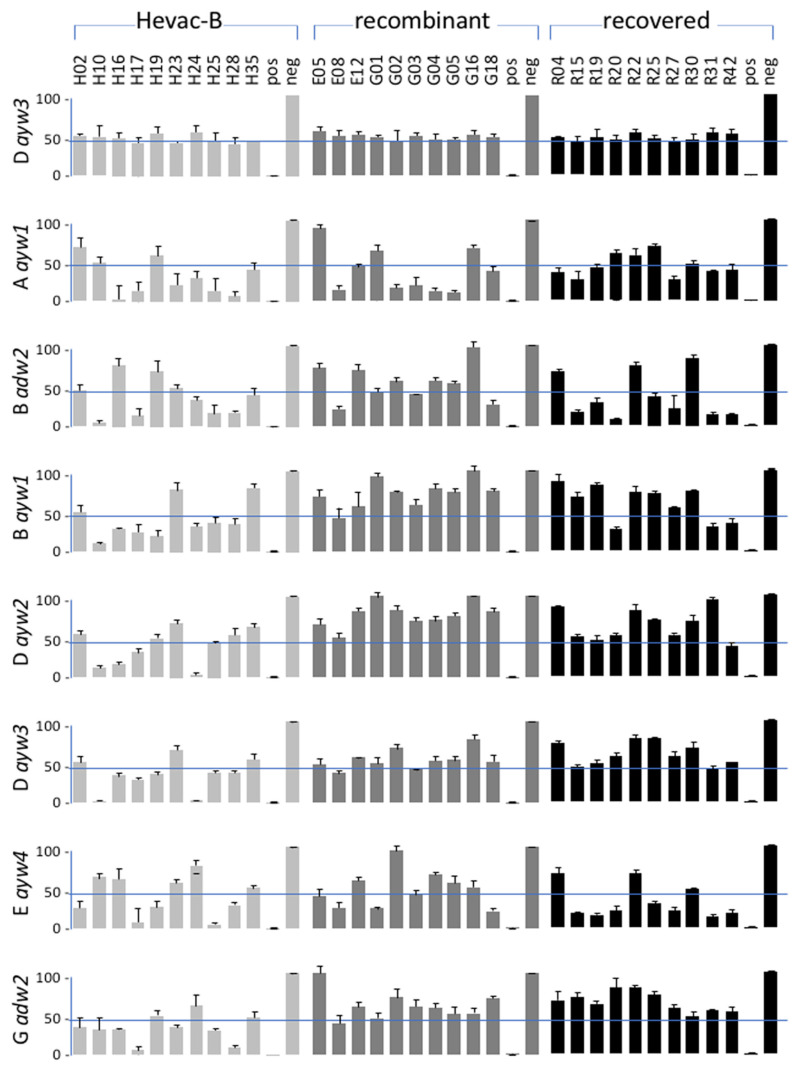
Analysis of vaccine-derived antibodies for their capacity to neutralize infectivity of virions coated with envelope proteins of different HBV genotypes/subtypes. Neutralization assay was performed using virions specific of the HBV envelope proteins of genotypes/subtypes, as indicated on the left. Virions were pre-incubated with antibodies from 10 Hevac-B vaccinees, 10 recombinant vaccine recipients, and 10 individuals who recovered from infection. Each IgG antibody was used at the IC50 concentration that is defined in Figure 4. Top histograms show the results of infection in percent of infection in the presence of anti-HBs-negative control (neg) with virions bearing the envelope proteins of the HBV genotype D ayw3 reference. Histograms below show the results obtained with inoculums containing virions coated with envelope proteins of the indicated genotypes/subtypes.

**Figure 5 vaccines-11-00791-f005:**
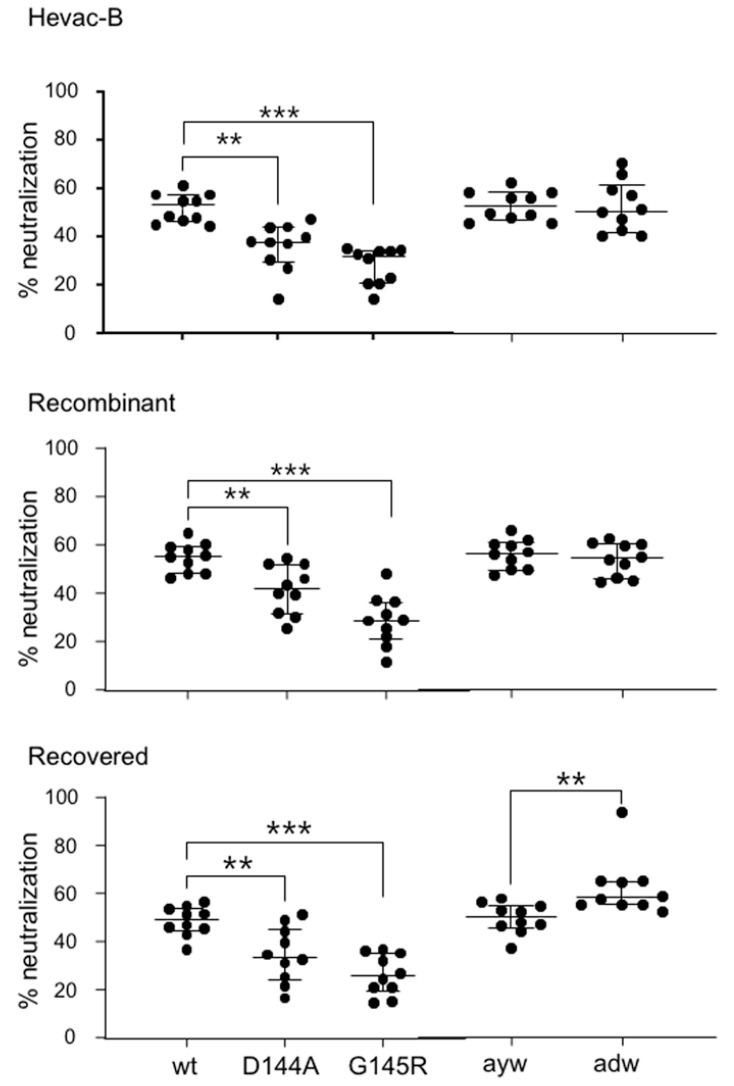
Anti-HBs antibodies are impaired in their capacity to neutralize HDV virions coated with envelope proteins of the main immune escape HBV variants D144A and G145. Antibodies from all groups could neutralize virions bearing envelope proteins of the adw and ayw subtypes. Each dot corresponds to an individual subject. Y-axis, percent neutralization. **, *p*-value < 0.01, ***, *p*-value < 0.001.

## Data Availability

Not acceptable.

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
