# Peer review of "Are International Units of Anti-HBs Antibodies Always Indicative of Hepatitis B Virus Neutralizing Activity?"

_vaccines, 2023, doi:10.3390/vaccines11040791_

Round 1

Reviewer 1 Report

This manuscript reports a very significant issue, namely, the efficacy of the various kinds of hepatitis B virus (HBV) vaccine preparations can not be reliably determined by the titer of anti-HBs antibody.  Instead, in vitro virus neutralization assay should be included in the quality control, and vaccine genotype/subtype should be matched with that of the circulacting HBV.  Furthermore, pre-S1 and pre-S2 antibody did not contribute to virus-neutralization titer. All of these points are important issues that have been neglected previously.  The data are robust and  clear-cut.  The presentation is logical and clear.  My only concern is that of virus neutralization test, which employed hepatitis delta virus (HDV) as a reporter.  This may have introduced an artificial requirement for the HBV virus replication assay.  This potential artifact should be ruled out.

Author Response

Q: This manuscript reports a very significant issue, namely, the efficacy of the various kinds of hepatitis B virus (HBV) vaccine preparations can not be reliably determined by the titer of anti-HBs antibody.  Instead, in vitro virus neutralization assay should be included in the quality control, and vaccine genotype/subtype should be matched with that of the circulating HBV.  Furthermore, pre-S1 and pre-S2 antibody did not contribute to virus-neutralization titer.  All of these points are important issues that have been neglected previously.  The data are robust and clear-cut. The presentation is logical and clear.  My only concern is that of virus neutralization test, which employed hepatitis delta virus (HDV) as a reporter.  This may have introduced an artificial requirement for the HBV virus replication assay.  This potential artifact should be ruled out.

R: The HDV experimental model was used as a surrogate for HBV for practical reasons. However, the HDV model has also the advantage of limiting our analysis to the capacity of antibodies to neutralize virus infectivity, a characteristic of antibodies induced by a sterilizing vaccine. This is reiterated throughout the text (lines 335-8 and 365-8). In addition, neutralization of infectivity should not differ between HBV and HDV because both virions bear the very same HBV envelope proteins, including the pre-S1, pre-S2 and HBs antigens.

We agree with the reviewer's suggestion in that, in vivo, there is more to the activity of vaccine induced antibodies than neutralization of virus infectivity (lines 335-8). There are additional aspects of the immune response to vaccination that we did not explore, such as Fc receptor-mediated anti-viral activities of antibodies, including their activity on antigen presenting cells, antibody-dependent cellular cytotoxicity and cell-mediated virus inhibition, which, all together play a role in preventing, or containing infection.

Reviewer 2 Report

I think this Manuscript address an important and pertinent problem. I think the Authors are right, that antibodies should be tested by an in vitro infection neutralization assay. The problem I see with this, is that such an assay can only be set up by specialist laboratories. I think the Authors need to address this aspect and write a section on how the neutralization assay can be implemented. I think what I am trying to say is that this assay cannot be run on a machine that is readily available. It require laboratory grown virus and skilled labour to run the neutralization tests. If possible, add a section as to how the Authors envision the test could be formulated or implemented so that it can reach a wider group of laboratories. But overall, I think the data presented is extremely sound and relevant. With the text itself, I only have two sentences, which I think need to me formulated in a different way to avoid confusion. In Abstract I would write "put more emphasis on" instead of "more focus should be payed". A few sections later I read the wording "sterilizing vaccine". I dont understand this use of the work "sterilizing". I maybe wrong, but I think it should read "sterile vaccine".

Author Response

Reviewer 2

Q: I think this Manuscript address an important and pertinent problem. I think the Authors are right, that antibodies should be tested by an in vitro infection neutralization assay. The problem I see with this, is that such an assay can only be set up by specialist laboratories. I think the Authors need to address this aspect and write a section on how the neutralization assay can be implemented. I think what I am trying to say is that this assay cannot be run on a machine that is readily available. It requires laboratory grown virus and skilled labour to run the neutralization tests. If possible, add a section as to how the Authors envision the test could be formulated or implemented so that it can reach a wider group of laboratories. But overall, I think the data presented is extremely sound and relevant.

R: We agree with the reviewer that testing for neutralization activity using an in vitro infection assay cannot be carried out as easily as running anti-HBs ELISAs, but as we already indicated in the discussion (lines 370-6), the robust and convenient assay can be standardized in a 96-well format, and carried out by pharmaceutical companies in the quality control procedures to test each lot of therapeutic material such as monoclonal antibodies or HBIG. The assay could also be conducted by virology reference laboratories for surveillance and prevention of HB vaccine failure. The latter sentence was added in the text (lines379-380).

Q: With the text itself, I only have two sentences, which I think need to me formulated in a different way to avoid confusion. In Abstract, I would write "put more emphasis on" instead of "more focus should be payed".

R: This sentence in the abstract has been modified to: "A greater emphasis should be placed on ensuring..." (lines 39-30).

Q: A few sections later I read the wording "sterilizing vaccine". I don't understand this use of the work "sterilizing". I may be wrong, but I think it should read "sterile vaccine".

R: The term "sterilizing vaccine" was indeed intended; it is the characteristic of a vaccine able to generate antibodies that bind to the incoming virus and, thereby, prevent viral entry into the host cells.